# Estimates of Crop Yield Anomalies for 2022 in Ukraine Based on Copernicus Sentinel-1, Sentinel-3 Satellite Data, and ERA-5 Agrometeorological Indicators

**DOI:** 10.3390/s24072257

**Published:** 2024-04-01

**Authors:** Ewa Panek-Chwastyk, Katarzyna Dąbrowska-Zielińska, Marcin Kluczek, Anna Markowska, Edyta Woźniak, Maciej Bartold, Marek Ruciński, Cezary Wojtkowski, Sebastian Aleksandrowicz, Ewa Gromny, Stanisław Lewiński, Artur Łączyński, Svitlana Masiuk, Olha Zhurbenko, Tetiana Trofimchuk, Anna Burzykowska

**Affiliations:** 1Remote Sensing Centre, Institute of Geodesy and Cartography, 02-679 Warsaw, Poland; katarzyna.dabrowska-zielinska@igik.edu.pl (K.D.-Z.); marcin.kluczek@igik.edu.pl (M.K.); anna.markowska@igik.edu.pl (A.M.); maciej.bartold@igik.edu.pl (M.B.); 2Space Research Centre, Polish Academy of Sciences, 00-716 Warsaw, Poland; ewozniak@cbk.waw.pl (E.W.); mrucinski@cbk.waw.pl (M.R.); cwojtkowski@cbk.waw.pl (C.W.); saleksandrowicz@cbk.waw.pl (S.A.); egromny@cbk.waw.pl (E.G.); stlewinski@cbk.waw.pl (S.L.); 3Statistics Poland, 00-925 Warsaw, Poland; a.laczynski@stat.gov.pl; 4State Statistics Service of Ukraine, 01601 Kyiv, Ukraine; s.masiuk@ukrstat.gov.ua (S.M.); o.zhurbenko@ukrstat.gov.ua (O.Z.); t.trofimchuk@ukrstat.gov.ua (T.T.); 5European Space Agency, 00044 Frascati, Italy; anna.burzykowska@esa.int

**Keywords:** crop monitoring, growing degree days, machine learning, satellite data

## Abstract

The study explores the feasibility of adapting the EOStat crop monitoring system, originally designed for monitoring crop growth conditions in Poland, to fulfill the requirements of a similar system in Ukraine. The system utilizes satellite data and agrometeorological information provided by the Copernicus program, which offers these resources free of charge. To predict crop yields, the system uses several factors, such as vegetation condition indices obtained from Sentinel-3 Ocean and Land Color Instrument (OLCI) optical and Sea and Land Surface Temperature Radiometer (SLSTR). It also incorporates climate information, including air temperature, total precipitation, surface radiation, and soil moisture. To identify the best predictors for each administrative unit, the study utilizes a recursive feature elimination method and employs the Extreme Gradient Boosting regressor, a machine learning algorithm, to forecast crop yields. The analysis indicates a noticeable decrease in crop losses in 2022 in certain regions of Ukraine, compared to the previous year (2021) and the 5-year average (2017–2021), specifically for winter crops and maize. Considering the reduction in yield, it is estimated that the decline in production of winter crops in 2022 was up to 20%, while for maize, it was up to 50% compared to the decline in production.

## 1. Introduction

The adaptation of agricultural monitoring systems to diverse geographical and geopolitical contexts presents unique challenges and opportunities. The complex task of transferring the EOStat crop monitoring system [1,2], initially developed and validated in the context of Polish agriculture, to the multifaceted landscape of Ukrainian agriculture is explored. The method relies on analyzing the anomalies in crop growth conditions in 2022 compared to the average patterns observed in previous years, using reference data from 2017 to 2021. Crop conditions are characterized by satellite vegetation indices, such as the normalized difference vegetation index (NDVI) [3,4,5], which reflect the greenness and vigor of crops; the land surface temperature (LST), which is associated with evapotranspiration and indicates water availability in the root zone; and agrometeorological parameters [6,7,8,9,10]. The primary objective is to provide timely and accurate assessments of crop production dynamics in Ukraine for the year 2022, leveraging the capabilities of Earth Observation (EO) data.

Ukraine holds a prominent position among the world’s leading agricultural producers and exporters, as noted by the United States Department of Agriculture (USDA). With its substantial contribution to the global market in products such as sunflower oil, sunflower hay, rapeseed, barley, corn, and wheat, Ukraine plays a crucial role in providing oilseeds and grains to meet global demand. This is particularly evident given Ukraine’s status as the world’s sixth-largest producer of corn, third-largest exporter of rapeseed, and seventh-largest wheat producer [11,12].

The significance of agriculture in Ukraine’s economy cannot be overstated. Arable land encompasses more than 55% of Ukraine’s total land area, underscoring the pivotal role of agriculture in sustaining the country’s economy. Agricultural products represent Ukraine’s most substantial exports, amounting to $27.8 billion in 2021, constituting 41% of the country’s total exports [13].

In light of ongoing hostilities and geopolitical uncertainties, the ability to monitor crop growth conditions and predict yields assumes critical significance for Ukraine. The EOStat system, with its capability to harness satellite data and agrometeorological indicators, emerges as a promising solution to address these challenges. Particularly in such conditions where the use of in-situ data collection is practically impossible, the development of solutions based on EO data becomes crucial.

A comprehensive methodology developed for adapting the EOStat system to the Ukrainian context is presented. The intricacies of satellite imagery analysis and machine learning techniques employed to forecast crop yields, with a particular focus on winter wheat, maize, and winter rapeseed, are explored. Additionally, insights into estimated crop yield anomalies for 2022 are provided, drawing on Copernicus Sentinel-1, Sentinel-3 satellite data, and ERA-5 agrometeorological indicators. The aim is to contribute to a more resilient and sustainable agricultural sector in Ukraine, despite the challenges posed by geopolitical uncertainties.

## 2. Materials and Methods

### 2.1. Administrative Division

Since 2020, Ukraine has functioned as a unitary state with three levels of administrative divisions: 27 regions (Figure 1), which include 24 oblasts (provinces), two cities with special status, and one autonomous republic. This administrative structure was introduced as part of reforms aimed at decentralizing governance and empowering local communities. Additionally, there are 136 raions, also known as ‘districts’, and 1469 hromadas, which are community-level administrative units. Before the year 2020, Ukraine was divided into 490 raions and 118 cities of regional significance, which were significant urban centers within their respective regions. The shift in administrative divisions reflects the evolving governance structure in Ukraine.

Throughout the course of this project, data at the region and raion levels are used to analyze and evaluate various aspects related to crop monitoring and yield prediction. These administrative divisions serve as meaningful units for assessing agricultural practices, crop performance, and regional variations. By examining the region- and raion-level data, a comprehensive understanding of the agricultural landscape in Ukraine can be obtained, enabling targeted analysis and informed decision-making.

### 2.2. Crop Yield Statistics

The State Statistics Service of Ukraine (SSSU) has demonstrated significant interest in the data derived from predictive models developed in the EOStat Poland project, specifically for implementation within the territory of Ukraine. Recognizing the value of these predictive models, the SSSU collaborated with the Institute of Geodesy and Cartography (IGiK) and the Space Research Centre of the Polish Academy of Sciences (CBK PAN) to exchange data and insights.

As part of this collaboration, the SSSU provided the IGiK and CBK PAN with comprehensive statistical data pertaining to yields and sowing areas of selected crops for the period of 2017–2021. This valuable dataset was made available in tabular format, offering information at two levels of detail: regions and raions (districts). The statistical data encompassed winter wheat, winter rapeseed, and maize yields, measured in decitonne [dt], reflecting the performance of these crops over the specified five-year period. A database of 7521 records was used in the work. Each record represented the yield of a selected crop in the year 2017–2021 for the district. The database contained data for winter wheat (2656), winter rapeseed (2304), and maize (2561). For each year (2017–2021), there were between 413 and 617 sample records for the crop.

To present a comprehensive overview of the available data, any gaps or limitations are visualized in Figure 2A for the regional level and Figure 2B for the district level. Notably, the dataset revealed that the least amount of in-situ data were obtained for winter rapeseed, which may introduce some limitations when assessing and analyzing the performance of this particular crop.

This collaborative effort between the SSSU, IGiK, and CBK PAN highlights the significance of data sharing and cooperation in advancing crop monitoring and predictive modeling initiatives. By leveraging the available statistical data, together with satellite-based information and advanced algorithms, more accurate and reliable forecasts can be developed to support agricultural decision-making in Ukraine.

In the subsequent stages of the project, a critical aspect under investigation is the comparison of average annual yields for each of the selected crops across different years. This comparative analysis aims to assess the performance and variations in crop yields over time. The average annual yield data for each crop is presented in Figure 3, which provides a clear visualization of the annual average yields for the entire country. It is evident from the data that, consistently across all years, maize exhibits the highest average yields among the selected crops. Conversely, winter rapeseed consistently demonstrates the lowest average yields.

### 2.3. Crop Mask—Crop Recognition

A map of crop was developed using a time series of Sentinel-1 data. A set of C-band, dual polarization (VH + VV) mode Single Look Complex Sentinel-1 products, which covered nine orbits (orbit numbers: 7, 21, 36, 65, 80, 94, 109, 138, and 167), were used. The orbits are partially overlapped. Images for the period from mid-February to the beginning of October were acquired at a frequency of 12 days—20 images per orbit.

The classification was carried out in three phases using a modified method proposed by Wozniak et al., 2022 [2]. In the first phase of the classification, the database of objects was constructed. Pre-processing of all images was realized through projection to the UTM system, mosaicking (one mosaic per acquisition date), and filtering using an enhanced Lee filter. Next, coherence matrices [14] were calculated, and the H/α decomposition for dual polarization (VV + VH) [15] data were applied. Six basic parameters were obtained: three real parts of the coherence matrix—T11, T12, and T22, and three parameters of the H/α decomposition—entropy (H), alpha (α), and lambda (λ). Elements of coherence matrices were rescaled [16]. A segmentation was performed based on the vector data of the extension of fields extracted from optical Sentinel-2 images and the time series of λ parameters. The mean values for the six basic parameters for each segment were calculated for all acquisition dates.

In the second phase, the classifier was developed. Firstly, the legend was built. Ten types of crops, which are the most common in at least one Ukrainian region, were selected for the classification: maize, spring barley, winter wheat, sugar beet, potato, soya bean, winter rapeseed, sunflower, other spring cereals, and other winter cereals. An additional class, “no crops”, was added to detect fields that were not sown. Next, training and testing datasets were prepared. Due to a lack of in-situ data from Ukraine, data from the Polish Southern-Eats regions was used as they have similar climatic conditions, especially at the beginning and end of the growing season. According to meteorological data, the most similar course of the growing season 2022 in Ukraine was the season 2019 in Polish regions adjacent to Ukraine, so this growing season was used to train and test the classifier. The process of classifier fitting was realized for reference data in the following way. Phenological indices were calculated using the six basic parameters: time series of Normalized Ratio of Coherence Matrix Elements, Normalized Time Ratios, Linear functions between local signal extremes (points where the signal power trend changes considerably) in time series. Then, iterative Random Forest classifier fitting was applied to select the optimal input parameters for classification.

In the third phase, the initial database of objects, which contained statistics on six basic parameters, was enriched by adding phenological indices, which were selected as the most useful for classification during the classifier fitting process. The fitted classifier for Polish test sites was applied to all orbits that covered Ukraine. As a result, we obtained nine partially overlapped maps of crops and nine maps of probabilities of class assignment given by the Random Forest classifier. In some cases, the crop type classified as a segment differed between two overlapping maps. In the original method, the crop with a higher probability was selected as the final one. In the case of Ukraine, a double-check procedure was applied. First, based on regional data for Ukraine for 2021, we established the differences in the probability of the presence of the crop in the administrative unit. Then, we analyzed the probability classification per orbit. Based on these two variables, the following criteria were used to establish the crop in the overlapped by two orbits segment: (1) the crop with the higher probability of classification and the higher probability of presence was fixed in the final map; (2) in the case of the first crop with a higher probability but less probable presence than the second one, the first crop was maintained if the difference in map probability was higher than the difference in the presence of crops. Applying the above-described approach, all segments had to have defined a crop because, in the training set, there were no “no crop” samples. However, some fields were not sown. Therefore, the final step in classification was to detect unsown fields. To recognize such fields, we used a time series of dairy Sentinel-3 images to calculate the Normalized Difference Vegetation Index (NDVI). We were searching for segments whose NDVI mean value in the period from the middle of May to the end of June was lower than 0.4 in the majority of images. These objects were reclassified into the “no crop” class.

The overall accuracy tested for the Polish test site was 79.31%. The F-scores of the crops for which the yields were estimated were the following: maize—0.80, winter wheat—0.78, winter rapeseed—0.95. To evaluate the accuracy of the estimation of the classification in Ukraine, the structures of crops within administrative units for the years 2022 and 2021 were compared. The difference in the share of a crop in the structure of crops did not exceed 3%.

### 2.4. Satellite and Agrometeorological Data Processing

Daily images from the Sentinel-3satellites were acquired from two orbits, covering the territory of Ukraine. The OLCI sensor provided scenes at a resolution of 300 m in the 681 nm and 865 nm channels, which were used to calculate the NDVI index. The SLSTR sensor, at a resolution of 1 km, was used to derive surface temperature (LST). The acquired data were processed by mosaicking, cloud masking, and reprojection to the UTM Zone 36N coordinate system (EPSG: 32636) using the ESA SNAP 9.0 Processing Toolbox. Historical satellite data from 2017–2021 and instantaneous data for the current growing season in 2022 were collected.

Agrometeorological data covering the period from 2017 to 2022, with a resolution of 0.25 × 0.25 degrees, were derived from ECMWF’s ERA-5 reanalysis, available at the Copernicus Climate Data Store. This included hourly data on the surface level of 2-m air temperature, total precipitation, solar radiation downwards, and volumetric soil water content at depths of 0–7 cm and 7–28 cm. The parameters were aggregated into daily means or sums using the Climate Data Operators (CDO). Additionally, minimum and maximum daily air temperatures were calculated from the hourly data.

Next, datasets derived from Sentinel-3 and ERA-5 were spatially aggregated for the administrative divisions of Ukraine, focusing on arable land. A crop mask was applied to define the arable land class, and the native temporal resolution of the products was maintained. For each product’s specific grid, an area fraction image was generated to determine the proportion of the grid area covered by arable land. A threshold was established to remove pixels where arable land covered less than 30% of the area. This crop image was used to compute aggregated values (mean) for administrative units, weighted by the share of arable land. The aggregation scheme assigned more weight to pixels with a higher fraction of arable land, resulting in a regional mean value. Additional weights were applied to account for the area covered by administrative unit boundaries within the aggregated pixel. Pixels fully covered by the administrative unit were given more weight, while those with only a small part within the unit received less weight. Consequently, a database of Sentinel-3 and ERA-5-based predictors at the native temporal resolution was developed for all administrative units. The satellite data with meteorological reanalysis were spatially merged into administrative units, while temporally they were aggregated on a basis with an increasing number of growing degree days in a 100-day step. A comprehensive list of input data, including satellite, agrometeorological, and ancillary materials used in the system, is presented in Table 1.

### 2.5. Computing Environment Preparation

The source code of the EOStat system has been migrated to the CREODIAS cloud computing platform. The migration process involved utilizing a virtual machine (VM) image, which was deployed as instances on the CloudFerro Creodias New WAW3-1 cloud server. Subsequently, the system was adapted to the network configuration of the HORIZON OpenStack cloud. Following that, the VM parameters, such as the disk space allocation for satellite data storage, were specified for computation across the entirety of Ukraine. These parameters included a 512 GB virtual machine, 3 TB HDD magnetic storage for satellite and processing data, 128 GB of Random Access Memory (RAM), and a virtual Centralized Processing Unit (vCPU) with 32 cores and 32 threads.

### 2.6. Generating a Data Archive of Growing Degree Days

To ensure comparability of vegetation indices from season to season, the indices were converted from conventional calendar time to thermal time. Thermal time, defined as the accumulation of daily temperatures above a crop-specific threshold, determines the growth stage achieved by a crop. By incorporating the indices into thermal time, it becomes possible to calculate index anomalies by comparing them to the average value of the index during the corresponding crop growth stage in previous years. This approach offers a significant advantage in describing crop growth.

Thermal time was calculated for a day (d) of the year as so-called growing degree days (GDD) from the daily maximum (T_x_) and minimum (T_n_) air temperatures for day i using a Formula (1):(1)$$GDD_{d}=\sum_{i=0}^{d}[\left(\frac{T_{x,i}+T_{n_{,}i}}{2}-T_{b}\right)\times \left[\underset{conditional}{\underbrace{\left(\frac{T_{x,i}+T_{n_{,}i}}{2}-T_{b}\right)>0}}\right]]$$
where T_b_ stands for base temperature: 5 °C for winter wheat and winter rapeseed, and 10 °C for maize. In addition, T_n,i_ was replaced by T_b_ if T_n,i_ < T_b_, and T_x,i_ was replaced by 30 °C if T_x,i_ > 30 °C. The conditional part of the equation equals 1 if the condition is met and 0 otherwise, which means that only positive values (of the mean temperature reduced by the base temperature) are summarized. Based on daily GDD values, all yield predictors were derived for GDD values ranging from 150 °C to 1200 °C, with a step of 150 °C.

Resampling of NDVI and LST to thermal time allowed the derivation of normalized indicators proposed by Kogan (1997) [9], such as Vegetation Condition Index (VCI) and Temperature Condition Index (TCI) defined as:(2)$${VCI}_{GDD}=100\times \frac{{NDVI}_{GDD}-{NDVI}_{min,GDD}}{{NDVI}_{max,GDD}-{NDVI}_{min,GDD}}$$
(3)$${TCI}_{GDD}=100\times \frac{{LST}_{max,GDD}-{LST}_{GDD}}{{LST}_{max,GDD}-{LST}_{min,GDD}}$$
where

GDD indicates the growing degree days (thermal time),

NDVI_GDD_—an instantaneous NDVI value,

NDVI_min,GDD_ and NDVI_max,GDD_—minimum and maximum NDVI recorded in the period 2017–2021 at a particular location for a given GDD, respectively. The definition of TCI_GDD_ follows the same logic as VCI_GDD_.

### 2.7. Development of Yield Prediction for Crops

The crop yield forecasting models for Ukraine were specifically designed to predict yields at the regional level, based on the administrative divisions prior to 2020. In order to establish reliable yield forecasts, we focused on three regionally important crops for the year 2022: winter wheat (e.g., Dnipropetrovsk, Kherson), winter rapeseed (e.g., Mikolayiv, Ternopil), and maize (e.g., Cherkasy, Chernihiv).

For each administrative division region, the input array for crop yield forecasting consisted of an *r × c* matrix, where r represents the number of years for which predictors and reference crop yields were available, and c indicates the number of predictor columns.

The predictors included the following variables, calculated at eight growing degree day (GDD) levels:minimum, maximum, and mean air temperature;solar radiation;accumulated solar radiation since 1 April;soil moisture at 0–7 cm and 7–28 cm levels;total precipitation;accumulated precipitation since 1 April;Normalized Difference Vegetation Index (NDVI) based on GDD;Vegetation Condition Index (VCI) based on GDD;Land Surface Temperature (LST) based on GDD;Temperature Condition Index (TCI) based on GDD;one seasonal predictor, namely the maximum NDVI during the growing season.

The dimension r varied among crop types and administrative units. All predictors were scaled to the interval between zero and one. Highly correlated predictors with a correlation coefficient above 0.75 were removed. Furthermore, a feature selection procedure was applied. This procedure involved iteratively running eXtreme Gradient Boosting (implemented in the R package XGboost) to identify the predictors with the highest variable importance. The final eXtreme Gradient Boosting prediction model was trained using the selected predictors and crop yield residuals as the dependent variables.

To evaluate the model’s performance, cross-validation using the leave-one-out approach, which is a variant of k-fold cross-validation where k is equal to the number of observations in the dataset, was conducted. It is important to note that the feature selection process was repeated at each iteration to prevent the model from benefiting from the knowledge of the data used for validation. The model’s performance was assessed using various metrics, including the coefficient of determination (R^2^), which is the square of the correlation coefficient, as well as the mean absolute error (MAE) and root mean square error (RMSE).

## 3. Results

### 3.1. Growing Degree Days for Main Crops

Figure 4 displays the map of growing degree days for winter wheat and winter rapeseed at the end of July, while Figure 5 shows the GDD map for maize at the end of September. Figure 6 presents the GDD differences for winter crops compared to the previous year, 2021, and the 5-year average across regions. It is observed that the northern regions of the country have a shorter summer growing season, whereas the southern regions experience a longer growing season—GDD is higher. Notably, there are similarities in the duration of the growing season between the preceding year, 2021, and the current year, 2022. However, a discrepancy is evident when comparing with the 5-year average from 2017 to 2021, particularly in the middle and northern regions (Figure 6 and Figure 7).

### 3.2. Selected Predictors for Estimating Crop Yields by the ML Algorithm

Figure 8 depicts the spatial distribution of the primary predictors selected by the machine learning algorithm for estimating yields of winter wheat, winter rapeseed, and maize in 2022, respectively. Upon examining the maps, it became evident that both agrometeorological and satellite-based indicators played a significant role in the algorithm. However, for winter crops, there is a notable prevalence of agrometeorological predictors, such as 2 m air temperature (maximum and minimum), total precipitation, and surface solar radiation, in numerous regions. Conversely, for maize, the selected predictors primarily focused on volumetric soil water content and surface solar radiation in most regions.

The implemented crop yield forecasting system incorporates two types of data. One set of data comprises predictors that reflect the growth conditions of crops, as indicated by agrometeorological indicators. The other set of data consists of vegetation indices derived from satellite observations, which provide information about the current state of crops affected by these growth conditions. By combining both sets of predictors, the crop yield forecasting system achieves improved accuracy by mitigating the limitations of each group.

The agrometeorological data, obtained from climatological reanalysis or interpolation of synoptic observations, offer valuable information at a resolution of a few kilometers. This resolution is often sufficient for describing variables such as air temperature, insolation, or average precipitation across flat terrains. However, certain extreme weather events, such as heavy rainfall or winter frosts, cannot be adequately captured by broad-scale agrometeorological data alone. Hence, it becomes crucial to incorporate satellite-based predictors that capture the vegetation’s response to these adverse weather conditions. These satellite-derived indicators provide valuable insights into the actual state of crops affected by such extreme events.

At the same time, it is important to acknowledge the limitations of satellite vegetation indices. Firstly, when using medium-resolution imagery, individual pixels often represent a mixture of various crop types, introducing uncertainty. Secondly, the NDVI index can become saturated beyond a certain biomass level, limiting its sensitivity to higher crop yields [7,17]. Therefore, the combination of agrometeorological and satellite predictors proves beneficial for crop yield modeling as it leverages the strengths of both data sources while mitigating their respective limitations.

### 3.3. Crop Yields and Crop Losses

This section aims to provide an overview of crop yields and losses in Ukraine, focusing on the key factors influencing winter wheat, winter rapeseed, and maize production in regions. By examining the existing literature and available data sources, the methods aim to shed light on the state of crop yields in Ukraine and provide insights for future research and agricultural management strategies.

#### 3.3.1. Winter Wheat

The spatial distribution of winter wheat yield predictions across various regions as of the end of July (27th) depicts Figure 9. The highest yields were observed in the Kiev region—the region marked on Figure 1 as number (15), while Vinnytsia (10) exhibited notable yields. Unfortunately, due to insufficient datasets, it was not possible to determine yield predictions for the Kiev city region (16) and Sevastopol (23). A comparison was conducted between the estimated crop yields for the years 2022 and 2021. Substantial changes in yield loss, exceeding a 20% difference, were observed in regions such as Zaporizhia (24), Kiev (15), Odessa (11), and Ternopil (3) (as depicted on the left side of Figure 10). In order to gain a broader perspective, the complete dataset provided by the State Statistics Service of Ukraine (SSSU) from 2017 to 2021 was analyzed. This comprehensive analysis revealed that yield loss changes exceeding 20% were not confined solely to the regions of Zaporizhia (24), Kiev (15), Odessa (11), and Ternopil (3). Similar changes were also observed in neighboring regions such as Dnipropetrovsk (20), Kirovohrad (13), Mykolaiv (12), Kherson (21), Cherkasy (14), and Volyn (5) (as illustrated on the right side of Figure 10).

#### 3.3.2. Winter Rapeseed

Analysis revealed a diverse predictive pattern in winter rapeseed yields at the end of July 2022 (Figure 11). The Ivano-Frankivsk (2) and Vinnytsia (10) regions exhibited the highest yields of winter rapeseed, while the Volyn region (5) was predicted to have the lowest yields. Unfortunately, insufficient data hindered us from determining yield predictions for the Kiev city region (16) and Sevastopol (23). To gain further insights, we compared the yield structure of winter rapeseed in 2022 with the previous year (Figure 12). Significant changes in yield loss were identified in the Ivano-Frankivsk (2) and Vinnytsia regions (10), while positive changes were observed in the Volyn region (5) (as demonstrated on the left side of Figure 12). Moreover, when considering all available data from previous years, notable changes in the yield structure were noted in the Volyn (5) and Khmelnytskyi (8) regions. Additionally, loss changes were observed in the Kharkiv (26) and Ivano-Frankivsk (2) regions (as illustrated on the right side of Figure 12).

#### 3.3.3. Maize

The analysis of maize yields across regions in Ukraine revealed a consistent spatial pattern of high maize yields (as depicted in Figure 13). The eastern, western, northern, and southern regions of Ukraine exhibited high maize yields, with the Ternopil (3) and Vinnytsia (10) regions standing out with yields exceeding 90 tons per hectare. Unfortunately, due to inadequate data, yield predictions could not be made for the Kiev city region (15) and Sevastopol (23). When comparing the maize crop structure to the previous year (2021), significant changes in yield patterns were observed in the Donetsk region (25), indicating substantial yield losses (as shown on the left side of Figure 14). However, when considering the entire available dataset, positive differences in the maize crop structure were also observed in the Odessa region (11) (as depicted on the right side of Figure 15).

### 3.4. Evaluation of the Model Performance

Figure 15 and Figure 16 present a detailed year-by-year analysis of the performance of crop yield forecasts for winter wheat, winter rapeseed, maize, and all three crops combined. The highest accuracy in the end-of-season crop yield predictions was observed for maize (R^2^ = 0.83 in 2018 and R^2^ = 0.7 in 2019) and winter wheat (R^2^ = 0.73 in 2018 and R^2^ = 0.66 in 2019). However, the predictions for winter rapeseed exhibited lower accuracy (R^2^ = 0.58) (Figure 16). The overall coefficient of determination (R^2^) for all three crops ranged from 0.66 to 0.86. Nonetheless, it should be noted that these errors may be larger for crops and years with limited input data, particularly when considering the case of rapeseed cultivation. Figure 16 illustrates the variation in forecasting quality metrics across years, particularly the tendency to overestimate crop yields for all three crops in 2020.

The high performance of winter wheat yield forecasts in Ukrainian regions, with a root mean square error (RMSE) of approximately 6.0 t·ha*^−^*^1^, can be attributed to the low annual variability of yields, ranging from 40 to 60 dt·ha*^−^*^1^, which can be accurately approximated by the machine learning algorithm. Although the RMSE for rapeseed is similar to that of winter wheat, ranging from 4.8 t·ha*^−^*^1^ to 6.7 t·ha*^−^*^1^, the performance of winter rapeseed yield forecasts is relatively low, with a coefficient of determination ranging from 0.4 to 0.6. This can be attributed to the relatively small size of the training dataset compared to the crop yield data available for winter wheat and maize from the State Statistics Service of Ukraine. Additionally, due to significant data gaps in the reference data for 2017, yield forecasting for crops could not be estimated.

We chose to utilize XGBoost in our study for several reasons. These reasons include the characteristics of our dataset, which consists of five years of data, the suitability of XGBoost’s model architecture (decision trees), the nature of our structured data (vegetation indices, agrometeorological parameters, and crop yields), and the ease of interpreting feature importance scores provided by XGBoost [18]. While both XGBoost and deep learning models are machine learning algorithms, they differ in several key aspects. Firstly, the model architecture differs between XGBoost and deep learning models [18]. XGBoost employs an ensemble of decision trees as base learners, whereas deep learning models utilize artificial neural networks with multiple layers. Secondly, the input data format varies. XGBoost is well-suited for structured data arranged in rows and columns, while deep learning models have the capability to handle both structured and unstructured data, such as images and text.

Thirdly, computational requirements differ between the two approaches. Deep learning models typically demand more computational resources, such as GPUs or TPUs, and may require longer training times compared to XGBoost. Fourthly, XGBoost provides a higher level of interpretability compared to deep learning models. It can generate feature importance scores and decision rules, enabling a clearer understanding of the model’s decision-making process. On the other hand, deep learning models are often considered “black box” models, making it challenging to interpret and comprehend their predictions. Lastly, XGBoost has demonstrated good performance on small datasets, while deep learning models usually require larger amounts of data to achieve optimal results.

Considering these factors, we determined that XGBoost was a suitable choice for our study due to its compatibility with our dataset, its interpretability, and its ability to effectively handle structured data.

The prediction results obtained using the aforementioned methods were compared with the actual wheat and maize crop yields, thanks to the involvement of the State Statistical Service of Ukraine (as of the end of 2022). The maps below (Figure 17) depict the Mean Absolute Percentage Error (MAPE) of the predictions. The choropleth maps illustrate the model’s tendency to exhibit higher predictive errors on the eastern side of Ukraine. Conversely, in the western regions, these errors are noticeably smaller, averaging around 5% for both wheat and maize.

## 4. Discussion

In recent years, there has been a growing interest in the development and application of innovative techniques for crop yield forecasting in Ukraine [19,20]. These techniques encompass a wide range of approaches, including remote sensing technologies, statistical models, and machine learning algorithms. Remote sensing techniques, such as satellite imagery and vegetation indices, enable the monitoring of crop growth patterns, vegetation health, and spatial distribution, providing valuable information for yield estimation. Statistical models leverage historical yield records and meteorological data to establish relationships between yield and various factors, enabling predictions based on past trends. Machine learning algorithms offer the capability to process large volumes of data, identify patterns, and make accurate predictions by leveraging complex algorithms and predictive analytics.

In the literature, manuscripts related to remote sensing analyses in Ukraine can be found. Specifically, Kolotii et al. (2015) [21] conducted a study that is relevant to this field and highlights the findings of a study that aimed to compare the performance of different predictors in forecasting wheat yield in Ukraine. The study specifically focused on evaluating the effectiveness of biophysical indicators and satellite-derived predictors in capturing wheat yield variations. This research utilized a comprehensive dataset that included biophysical parameters such as leaf area index (LAI), vegetation indices (e.g., NDVI), temperature, and precipitation, along with satellite data from multiple sensors. Statistical analyses, including correlation analysis and regression modeling, were employed to assess the relationships between the predictors and wheat yield. The results indicated strong associations between certain biophysical parameters and wheat yield. For instance, the leaf area index was found to be a significant predictor, indicating that higher LAI values were generally associated with higher wheat yields. Similarly, vegetation indices derived from satellite data, such as the normalized difference vegetation index (NDVI), demonstrated a positive correlation with wheat yield. Furthermore, the research examined the influence of weather variables, including temperature and precipitation, on the forecasting of wheat yield. These studies align with the findings of the present study. The results indicated that weather parameters, especially during critical growth stages, significantly influenced the variability of wheat yields.

The Joint Research Centre (JRC) has also conducted research on monitoring crop conditions in areas related to Ukraine [22]. The study, carried out by the JRC, focuses on analyzing and assessing the agricultural outlook and crop conditions in Ukraine, taking into account the broader European neighborhood. The research aims to provide valuable insights into the state of crop health and development in Ukraine, considering various factors that influence agricultural outcomes. By examining and evaluating the crop conditions, the JRC aims to contribute to a comprehensive understanding of the agricultural landscape in Ukraine and support evidence-based decision-making in the region. The results obtained in this study are consistent with the cited manuscript. The authors noted a decrease in yields of the main crops such as wheat, rapeseed, and barley in the eastern regions, while an increase in wheat yield of up to approximately 100% was observed in the Zakarpats’ka region (1). When analyzing the conditions related to rapeseed cultivation, the highest growth compared to the five-year average was recorded in Ivano-Frankivs’ka (2), reaching around 76%. These findings confirm the trends and variations in crop yields as reported in the referenced manuscript.

In addition, Franch et al. (2019) [23] conducted analyses on the territory of Ukraine. Their study also introduced a novel crop yield model that utilized the Difference Vegetation Index (DVI) derived from Moderate Resolution Imaging Spectroradiometer (MODIS) data at 1 km resolution. Similar to the aforementioned study, the model employed un-mixing techniques to isolate a pure wheat signal within the pixel, allowing for precise yield assessment. The model was then applied to estimate winter wheat yield at both national and subnational levels in the United States and Ukraine for the period spanning from 2001 to 2017. In the case of similar studies of yield prediction [24] in a climatic zone in Poland, predictions were obtained for wheat on the basis of MODIS data (LAINDVI and LST) and meteorological data, reaching values of R*^2^* = 0.89 and RMSE = 6.1–8.55.4 dt/ha [25], which shows a particularly strong alignment with the results obtained for 2018. Of particular relevance are the results on food security, where accurate estimates are essential for supply chains to deliver food where it is needed. Of particular relevance are the results on food security, where accurate estimations are essential to enable supply chains to supply food and make data-driven decisions for policymakers [26,27].

## 5. Conclusions

This study presents the results of crop yield forecasting during the growing season for winter wheat, winter rapeseed, and maize in regions of Ukraine. While official statistics were not available for all twenty-seven regions, leading to spatial gaps in the predictions at this level, the performance of the forecasts was found to depend on the size of the training dataset, specifically the length of the time series of reference yield statistics, and the reliability of satellite-based vegetation and meteorological products.

The findings from the EOStat System for Ukraine highlight the importance of integrating satellite and climatological data for accurate modeling of crop yields. In addition to providing end-of-season forecasts that can be used by Statistical Service Units for official assessments, in-season forecasts enable early warning in the case of unfavorable growth conditions.

Furthermore, despite the relatively short time-series data of only five years used for modeling crop yields in Ukraine, this study confirms the effectiveness of remote sensing techniques in supporting crop monitoring efforts. The fusion of satellite data with climatological information has demonstrated its potential for accurate crop yield forecasting and early detection of adverse conditions.

Finally, the results of this research suggest that the model developed for Poland can certainly be utilized and adapted for neighboring countries and regions. This observation has created new challenges that allow for a broader perspective on crop prediction, not only in Poland but also in regions with similar agrometeorological conditions worldwide. By enabling the adaptation of this model to different locations, tailored tools and strategies can be developed for other countries, such as Ukraine. This opens up new prospects for the application of remote sensing techniques in agriculture, contributing to the advancement of knowledge regarding the optimal utilization of resources and crop prediction in various regions of the world.

## Figures and Tables

**Figure 1 sensors-24-02257-f001:**
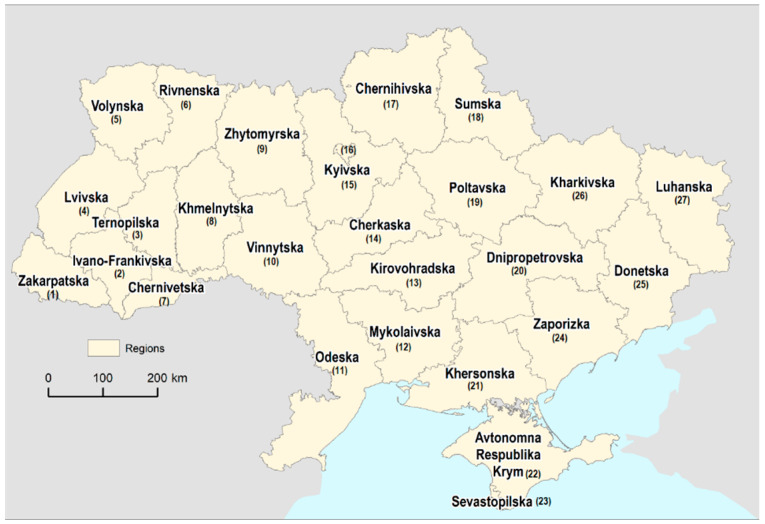
Regions in Ukraine in 2022.

**Figure 2 sensors-24-02257-f002:**
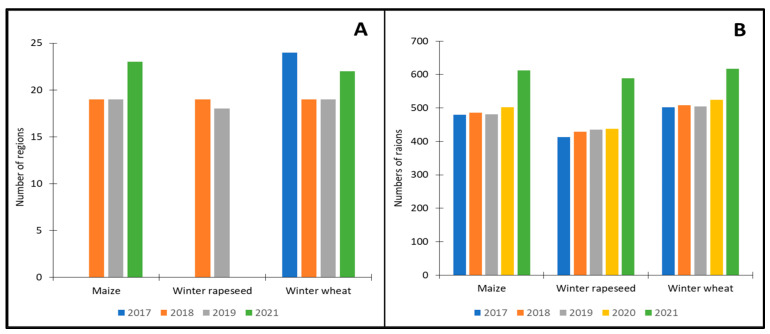
Number of regions (**A**) and number of raions (**B**) for which in-situ data were available by crop and year (total number of regions: 27).

**Figure 3 sensors-24-02257-f003:**
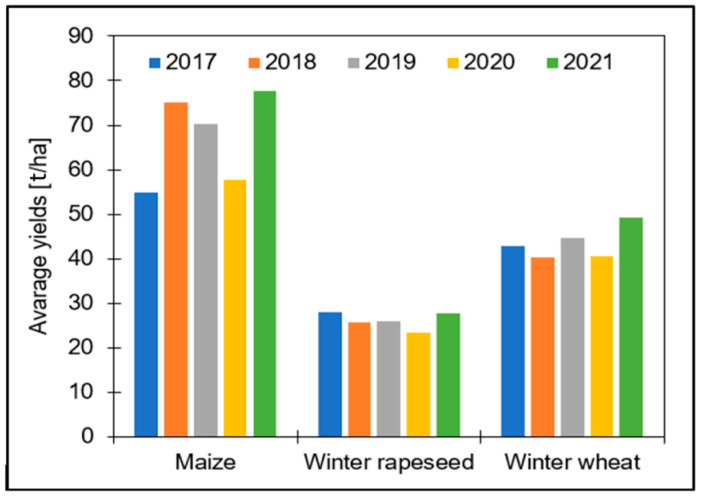
Average yields from in-situ data.

**Figure 4 sensors-24-02257-f004:**
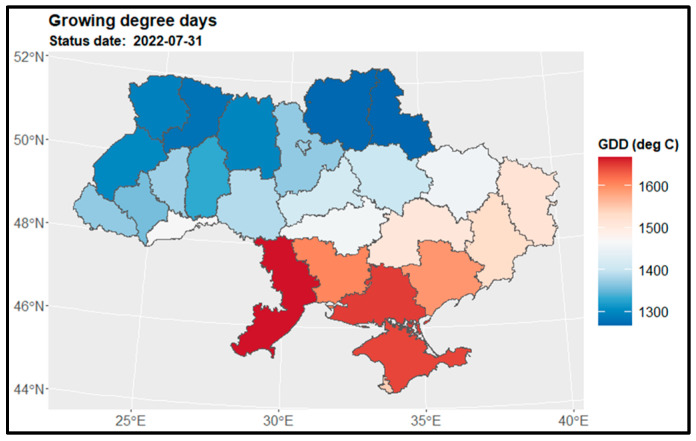
Spatial distribution of growing degree days for winter wheat and winter rapeseed at the end of July 2022 over regions.

**Figure 5 sensors-24-02257-f005:**
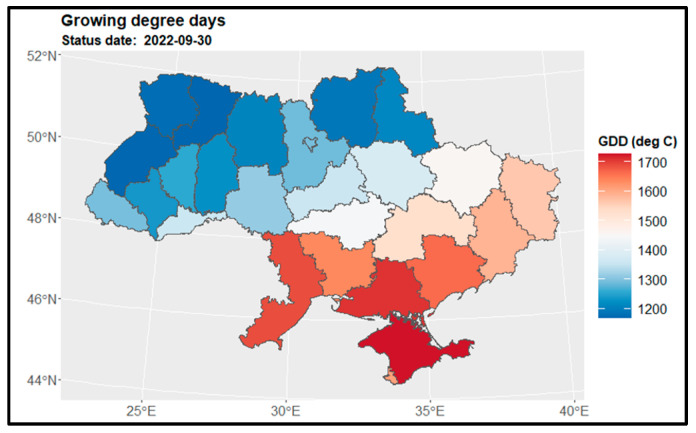
Spatial distribution of growing degree days for maize at the end of September 2022 over regions.

**Figure 6 sensors-24-02257-f006:**
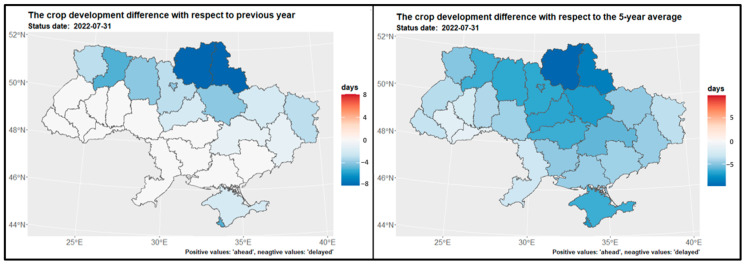
Spatial distribution of GDD differences at the end of July for winter crops regarding the previous year 2021 (**left**) and the 5-year average 2017 2021 (**right**) over regions.

**Figure 7 sensors-24-02257-f007:**
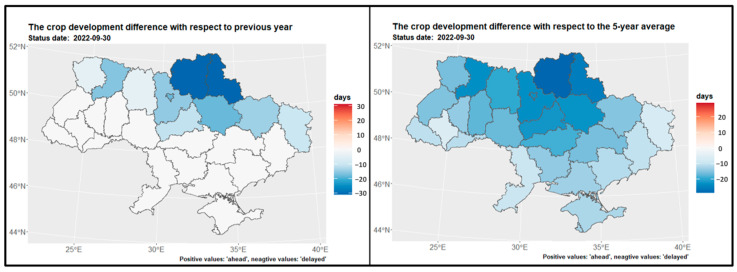
Spatial distribution of GDD differences at the end of September regarding the previous year 2021 (**left**) and to the 5-year average 2017 2021 (**right**) over regions.

**Figure 8 sensors-24-02257-f008:**
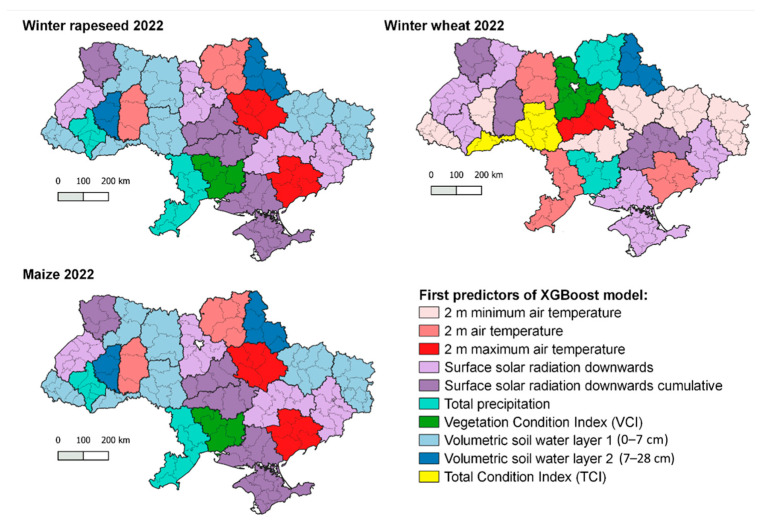
Map of first predictors of the XGBoost model for winter rapeseed, winter wheat, and maize in 2022 for regions.

**Figure 9 sensors-24-02257-f009:**
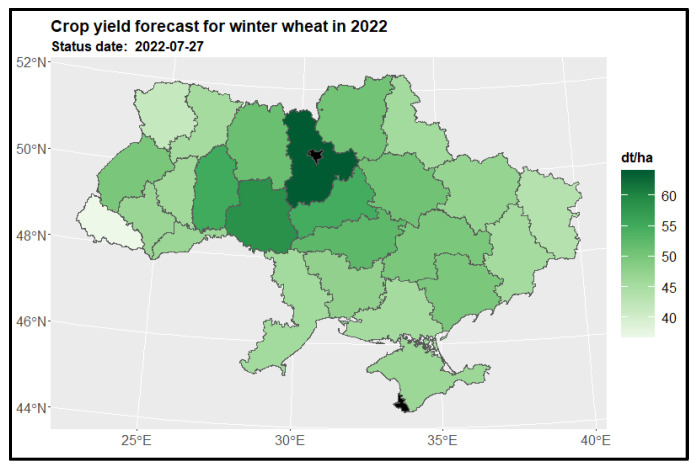
Crop yield forecast for winter wheat in 2022 for regions.

**Figure 10 sensors-24-02257-f010:**
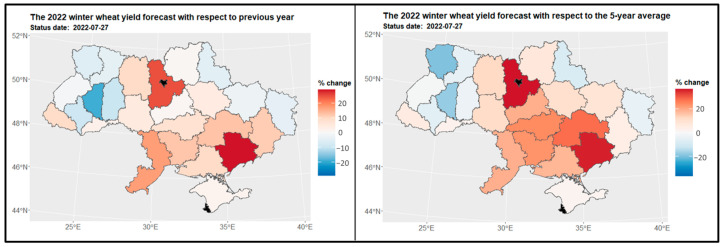
Map of the change in winter wheat yield in 2022 compared to the previous year for regions (on the **left side**) and to the 5-year average (on the **right side**) for regions. Areas in red indicate a decrease in yield, where -20% represents a yield that is 80% of the reference period’s yield, and 20% indicates an increase to 120% of the reference period’s yield.

**Figure 11 sensors-24-02257-f011:**
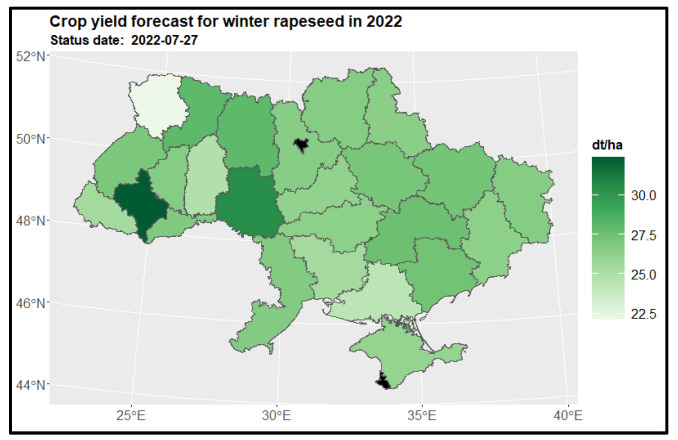
Crop yield forecast for winter rapeseed in 2022 for regions.

**Figure 12 sensors-24-02257-f012:**
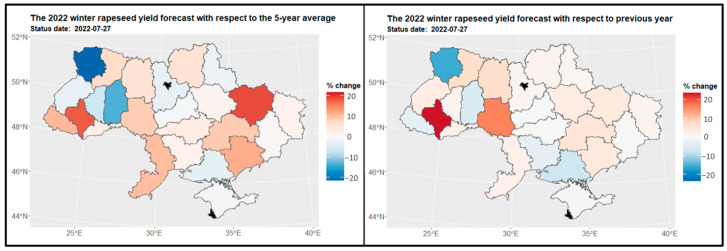
Map of the change in winter rapeseed yield in 2022 compared to the previous year for regions (on the **left side**) and to the 5-year average (on the **right side**) for regions. Areas in red indicate a decrease in yield, where −20% represents a yield that is 80% of the reference period’s yield, and 20% indicates an increase to 120% of the reference period’s yield.

**Figure 13 sensors-24-02257-f013:**
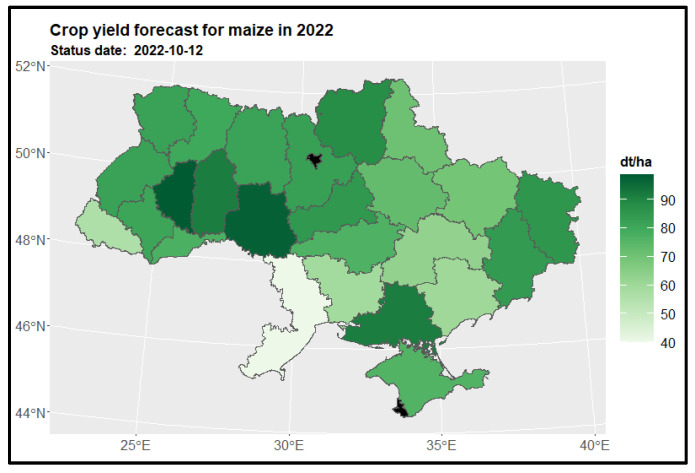
Crop yield forecast for maize in 2022 for regions.

**Figure 14 sensors-24-02257-f014:**
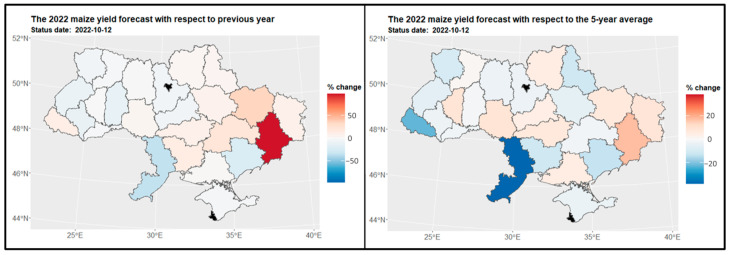
Map of the change in maize yield in 2022 compared to the previous year for regions (on the **left side**) and to the 5-year average (on the **right side**) for regions. Areas in red indicate a decrease in yield, where −20% represents a yield that is 80% of the reference period’s yield, and 20% indicates an increase to 120% of the reference period’s yield.

**Figure 15 sensors-24-02257-f015:**
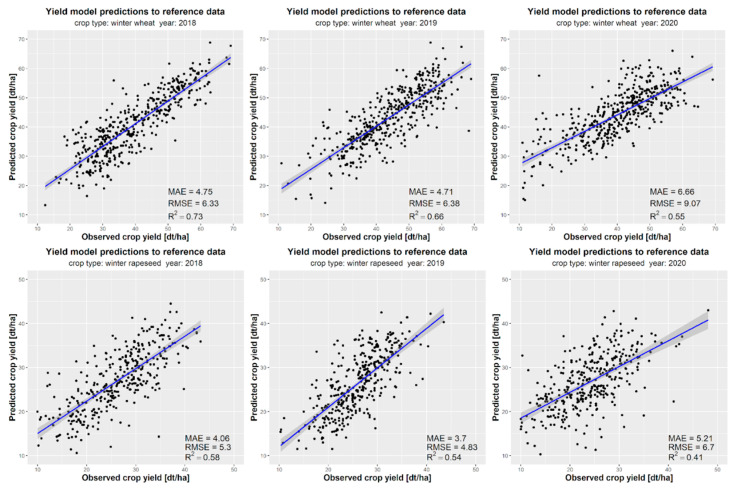
Comparison of model prediction results with reference data at regions for winter wheat (**first row**), winter rapeseed (**second row**), and maize (**last row**) in the period 2018–2020.

**Figure 16 sensors-24-02257-f016:**
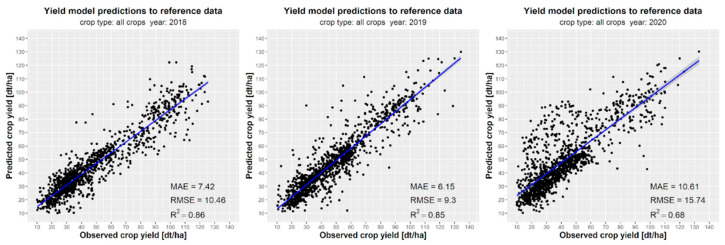
Comparison of model prediction results with reference data for regions of all analyzed crops (maize, winter wheat, and winter rapeseed) for the 2018–2020 years.

**Figure 17 sensors-24-02257-f017:**
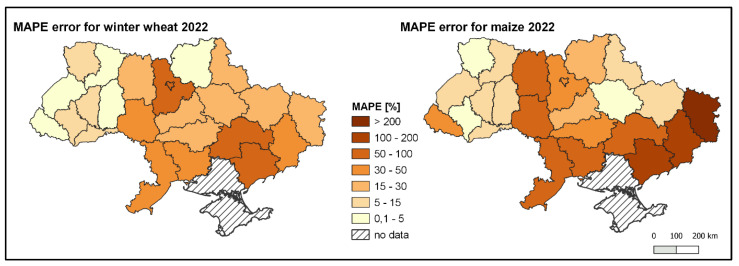
MAPE error for winter wheat (**left**) and maize (**right**) yield in Ukraine in 2022.

**Table 1 sensors-24-02257-t001:** Specification of satellite, agrometeorological, and ancillary data used to derive crop yield predictors.

Name	Source	Temporal Resolution	Spatial Resolution
Satellite indices			
NDVI	Sentinel-3	1 day	300 m
LST	Sentinel-3	1 day	1000 m
Ageometeorological parameters			
Air temperature	ERA-5	1 h	0.25 degree
Precipitation	ERA-5	1 h	0.25 degree
Surface radiation	ERA-5	1 h	0.25 degree
Soil moisture	ERA-5	1 h	0.25 degree
Crop mask			
Crop classification	CBK PAN (Wozniak et al., 2022 [2])	static	polygons
Administrative units			
Regions, raions	Data before 2020: www.ukrmap.com.ua accessed on 22 March 2024 Data since 2020: Data.humdata.org accessed on 22 March 2024	static	polygons

## Data Availability

Crop prediction data will be made available upon request.

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
