# Peer review of "Estimates of Crop Yield Anomalies for 2022 in Ukraine Based on Copernicus Sentinel-1, Sentinel-3 Satellite Data, and ERA-5 Agrometeorological Indicators"

_sensors, 2024, doi:10.3390/s24072257_

Round 1
Reviewer 1 Report
Comments and Suggestions for Authors
The article is interesting; however, it needs changes for its publication.
Keywords must not be in the title, so the number of keywords can be reduced or those in the title must be changed.
The "Introduction" section was not written as an Introduction to a scientific article. It does not present the problem, the rationale, the innovative character, and, mainly, the general and specific objectives of the work, comprehensively and interestingly. In fact, and erroneously, from line 44 onwards, it deals with methodology. Ask yourself: what are the challenges of using the EOStat System in Ukraine, compared to its already validated use in Poland? What is the benefit of applying this system to Ukrainian agriculture? The Introduction, therefore, must be completely rewritten.
For the "Material and Methods" section, the information presented in 2.3 does not qualify as a Methodology. In fact, his interpretation would serve as a basis for exposing much about the justification of the work in the Introduction. In Table 1, the term "Glocal" should be corrected.
What is the size, in fact, of the training series and what is the size of the validation series, of the prediction model? This was not clear from the methodology. Wouldn't the training and validation series be too small? Isn't this what led to the high RMSE values? What is the consequence of these high RMSE values for the applicability of the proposed forecasting system?
The conclusion is more of a discussion of the results than a conclusion itself, which may have happened due to the lack of a clear definition of the objectives. The discussion provided in the "Conclusion" section is interesting, but it should be presented somewhere else.
Comments on the Quality of English Language
There are some typos throughout the text. Please, check it again.
Reviewer 2 Report
Comments and Suggestions for Authors
Overall, the manuscript is well written, it is the adaptation of a Polish crop yield system to the Ukraine situation. The text could be improved by reducing the length to about 15 pages and including a flow chart of the methodology would help the reader to better understand the flow of data and manipulations to produce the outcome.
As Sensors is a journal focusing on remote sensing sensors, I would argue that the authors could elaborate upon the integration of radar and optical sensors in this article.
Introduktion
- All abbreviations should be spelled out.
- line 22 hectares, for such large numbers use square kilometers.
- line 47 NDVI, OLCI, SLSTR etc. please spell out.
- Materials and methods
Numbers 1 – 10 should be spelled e.g. 1 should be one, 2 two etc., 11 is 11 and so forth.
Figure 1, and all figures thereafter, the resolution is insufficient, details cannot be distinguished.
Figure 2, please remove, limited information content.
Figure 6, the resolution is low and images small, so it gives only very limited insight, for example rapeseed and soybeans cannot be seen.
Line 207, 9 change to nine
line 208, what is the adjust crop type?
Figure 7, very difficult to see anything, resolution has to much higher, perhaps resample data?
Results and discussion
Split this into Results and Discussion, this will help the reader a lot.
Figure 8, the name of the crop is not mentioned in the caption. Same for figures 8, 10 and 11.
Reviewer 3 Report
Comments and Suggestions for Authors
The authors estimated the crop yield in 2022 based on agro-meteorological and satellite remote sensing indicators. The recursive feature elimination method and extreme gradient boosting regressor were employed to forecast crop yields. The research is interesting and the results are solid. I suggest the editors that only minor revision is required before acceptance.
Specific comments:
1.In the introduction section, the authors are suggested to add the recent references on the methods of “recursive feature elimination method and Extreme Gradient Boosting regressor”.
2. In the figure 12, it seems that the major predictors of XGBoost model are agro-meteorological indicators rather than remote sensing VCI and TCI. The title of the paper is “based on Copernicus Sentinel-1, Sentinel-3 satellite data and ERA-5 agro-meteorological indicators”, which is more significant for yield prediction in this paper, agrimeteorology or remote sensing indicators?
3. In the figure 14, the caption is not clear. Is it “the lost winter wheat yield” or “the 2022 winter wheat yield forecast” in the figure? I suggest authors maintain them consistently. For the legend, did the red color in the map mean yield increase? 20 or -20 means 120% and 80%?
Round 2
Reviewer 2 Report
Comments and Suggestions for Authors
No further comments